Resistance exercise interventions for sarcopenia and nutritional status of maintenance hemodialysis patients: a meta-analysis

http://orcid.org/0000-0002-6569-6008 Li Li 1 2 1311016216@qq.com
Ma Xiaolan 1 3
Xie Chunyan 2
Li Yamin 2 aminny@csu.edu.cn
1 Department of Urology, The First Affiliated Hospital of Xinjiang Medical University , Urumqi, Xinjiang , China
2 Clinical Nursing Teaching and Research Section, The Second Xiangya Hospital, Central South University , Changsha, Hunan , China
3 School of Nursing, Xinjiang Medical University , Urumqi, Xinjiang , China
Capusa Cristina
Electronic publication date: 2024 Feb 5
Publication date: 2024
Volume: 12
Electronic Location ID: e16909
Received 2023 Oct 9; Accepted 2024 Jan 17
Copyright: © 2024 Li et al.
Copyright year: 2024
Copyright holder: Li et al.
License: This is an open access article distributed under the terms of the Creative Commons Attribution License, which permits unrestricted use, distribution, reproduction and adaptation in any medium and for any purpose provided that it is properly attributed. For attribution, the original author(s), title, publication source (PeerJ) and either DOI or URL of the article must be cited.
License URL: https://creativecommons.org/licenses/by/4.0/

Keywords: Resistance exercise, Hemodialysis patients, Sarcopenia and nutritional status

Funding: Major Scientific and Technological Projects in Hunan Province 2020SK2085 Natural Science Foundation of Xinjiang Uygur Autonomous Region 2018D01C196 This work was supported by the Major Scientific and Technological Projects in Hunan Province (grant number 2020SK2085) and the Natural Science Foundation of Xinjiang Uygur Autonomous Region (2018D01C196). The funders had no role in study design, data collection and analysis, decision to publish, or preparation of the manuscript.

==============================
Background

A previous meta-analysis showed that exercise training improves patients’ physical function and health status. However, the efficacy of resistance exercise (RE) in improving patients’ physical function and nutritional status is uncertain.

Objective

To evaluate the effect of RE on sarcopenia and nutritional indicators in maintenance hemodialysis (MHD) patients.

Design

A meta-analysis.

Methods

Randomized controlled trials up to March 28, 2023 were searched from eight databases, including PubMed, Web of Science, Embase, Cochrane, China National Knowledge Infrastructure, Wan Fang, China Science and Technology Journal Database, and CBM. The risk of bias of the literature eligible for inclusion was assessed using the Cochrane risk-of-bias tool. When a high heterogeneity was detected, a random-effects model was used. Egger’s tests were used to assess publication bias. This review was conducted in accordance with the PRISMA guidelines. The reliability of evidence was assessed using the Grading of Recommendations Assessment, Development and Evaluation method, and a meta-analysis of the collected data was performed using Review Manager 5.4.1 software.

Results

Nine studies that met the criteria were identified, and 541 patients were included in these research.

Subjects

The results of this review showed that RE improved patients’ grip strength levels (mean difference (MD) = 4.39, 95% confidence interval (CI) [3.14–5.64]; P < 0.00001), 6 min walking distance (MD = 40.71, 95% CI [8.92–72.49]; P = 0.01), muscle mass (MD = 4.50, 95% CI [2.01–6.99]; P = 0.0004), and serum albumin level (MD = 3.16, 95% CI [1.13–5.19]; P = 0.002) compared with the controls. However, the improvement caused by RE on hemoglobin (MD = 1.69, 95% CI [−1.49 to 4.87], P = 0.30) and cholesterol (MD = 2.33, 95% CI [−5.00 to 9.65], P = 0.53) levels was not statistically significant.

Conclusion

RE showed a significant effect on muscle function and strength of MHD patients. This meta-analysis provides new ideas on the efficacy of RE in muscle function and strength of MHD patients. The use of consistent RE patterns and nutritional interventions should be considered in future studies for further assessment of its effects. In the future, more high-quality studies will be required to verify these results.

Implications for practice

This meta-analysis identified the effect of RE on muscle strength, muscle function, and walking ability of HD patients, and provided a basis for clinical formulation of the optimal timing of intervention and the optimal frequency, intensity, modality, and content of intervention.

Patient or public contribution

No patient or public contribution because it does not apply to my work. This review has been registered at the International Platform of Registered Systematic Review and Meta-analysis (INPLASY) (registration number: INPLASY202340078).

Introduction

Maintenance hemodialysis (MHD) is the most common form of treatment for patients with chronic kidney disease (CKD) (Zhu, 2022). Patients receiving hemodialysis (HD) are limited by dialysis frequency, duration, and treatment modalities, and their physical activity level and exercise capacity are considerably reduced relative to normal individuals. Most dialysis patients develop sedentary lifestyle habits, which predispose them to manifestations of malnutrition, metabolic disorders, and sarcopenia (Wu, Hsu & Tzeng, 2022; Tayebi, Ramezani & Kashef, 2018; Li et al., 2021). Sarcopenia in MHD patients has a prevalence of myasthenia ranging from 13.7% to 73.5% (Dong, Zhang & Yin, 2019), and the prevalence of malnutrition ranges from 22.4% to 75% (Wu, Wang & Zhou, 2019). Malnutrition and myasthenia are common in MHD patients, and malnutrition is an essential cause of the development of myasthenia (Kurajoh et al., 2022). Myasthenia and malnutrition can cause patients to suffer from cognitive dysfunction, fractures, falls, metabolic disorders, increased protein energy consumption, and other adverse conditions. These conditions can also lead to reduced quality of life, increased length of hospital stay, mortality, and disability (Shu et al., 2022; Giglio et al., 2018; Gregg et al., 2021; Lee et al., 2020).

The current available treatment options regarding the nutritional status of HD patients are pharmacotherapy, dietary therapy, oral nutritional supplements, and exercise therapy. Relevant guidelines suggest that planned implementation of exercise rehabilitation is essential for the treatment of malnutrition in HD patients; exercise rehabilitation will increase their body metabolic rate, promote muscle synthesis and reduce muscle breakdown, improve their physical function and nutritional status, help patients with weight control, improve cardiovascular health, increase bone density, and reduce pain (Milam, 2016; Zhang, Ma & Zuo, 2021; Chan, Cheema & Fiatarone Singh, 2007). One meta report (Pu, 2018) stated that exercise can improve patients’ physical function and health status. However, the improvement in a patient’s nutritional status is still unclear. Resistance exercise (RE) can effectively prevent loss of muscle function; enhance patients’ muscle strength and mass, quality of life, and nutritional status of HD patients; and reduce protein energy consumption (Zelko et al., 2022; Deus et al., 2021; do Valle et al., 2020). However, our review revealed that the studies related to the effect of RE on the improvement of nutritional status and strength in this population are divided, with several research reporting improvements in the content of nutritional indicators (Yan, Zha & Peng, 2022; Cai et al., 2022; Chung, Yeh & Liu, 2017) and others offering no such finding (Pu, 2018; Dai & Ma, 2021). Whether RE can improve nutritional indicators in patients with MHD is unclear. Thus, this study mainly aimed to determine the relationship between sarcopenia in patients with MHD and nutritional status after exercise.

Materials and Methods

This review followed the PRISMA guidelines, PRISMA checklist, and Cochrane Handbook for meta-analysis (see Fig. 1). This study’s prospective protocol was registered at the International Platform of Registered Systematic Review and Meta-analysis Protocols (INPLASY: 202340078, DOI 10.37766/inplasy2023.4.0078).

Figure 1 Preferred Reporting Items for Meta-Analyses (PRISMA) flow diagram for the study selection process.

Eligibility criteria

The eligibility criteria followed the Population, Intervention, Control and Outcome study principles.

Participants

Patients on HD aged ≥ 18 years on dialysis treatment for ≥3 months.

Interventions

RE lasted for at least 8 weeks, and the exercise frequency was at least twice a week. Personal exercise must last at least 20 min. Exercise interventions can be intra- or inter-dialysis.

Controls

Usual care was used in the controls.

Outcomes indicators of interest

Sarcopenia-related indicators: muscle function (6 min walk test), muscle strength (grip strength).;

Nutrition-related indicators: hemoglobin (Hb), serum albumin (ALB), etc.;

Study type

We included randomized controlled trials (RCTs).

Exclusion criteria

We excluded literature that met the following conditions: (1) review literature; (2) non-RCTs; (3) literature that does not match the content of the current study; (4) literature from before 2000, literature for which detailed data, abstracts, and full text were unavailable.

Search strategy

Two investigators searched the literature independently in accordance with the search formula, and any disagreement on the included studies was discussed and resolved by a third investigator. Eight databases were searched: PubMed, Web of Science, Embase, Cochrane and China National Knowledge Infrastructure (CNKI), Wan Fang, VIP, and CBM. The search period was from creation to March 28, 2023.

Chinese search terms: maintenance hemodialysis/hemodialysis; resistance exercise/resistance training/resistance exercise/resistance training; English search terms: maintenance hemodialysis/maintenance dialysis/hemodialysis/dialysis/blood dialysis/MHD; resistance training/resistance exercise/resistive exercise/strength training/strength-type training/strength-type exercise/physical training (details of the search strategies are displayed in Supplemental File 1).

Study selection

After removing duplicate articles, two review researchers (L, L. and XL, M.) independently and sequentially screened potential studies through the search strategy and in accordance with the inclusion and exclusion criteria.

Data extraction

The extracted data included the following: (1) essential characteristics of the included studies, such as first author, year, and country; (2) sample size of the included studies; (3) type, intensity, frequency, and duration of intervention of the included studies; (4) corresponding outcome indicators of the included studies.

Two researchers (L, L. and XL, M) independently extracted data from the eligible literature using an agreed-upon form designed to record data. The authors were contacted for additional information if data could not be extracted. Any disagreements during data extraction were resolved with the support of a third author (YM L).

Literature quality evaluation and risk-of-bias assessment

Two researchers (XL, M and CY, X.) independently assessed the risk of bias for each eligible study according to the Cochrane Collaboration’s RCT criteria. We resolved differences by discussion or by involving other review authors.

We assessed the risk of bias based on the (1) generation of randomized sequences, (2) allocation concealment, (3) blinding of participants and staff, (4) blinding and outcome assessments, (5) incomplete outcome data, (6) selective outcome reporting, and (7) other biases. The items assessed were categorized as high risk, low risk, and unclear. The results of these questions were graphed and assessed using Review Manager 5.4.1, Egger’s tests were used to assess publication bias.

In addition, we assessed the quality of the included studies using the Jada scale, which comprised the following four entries: generation of random sequences, method of randomization concealment, implementation of blinding, and reasons for loss of data and withdrawal of the included studies for assessment.

Evaluation of the overall quality of evidence

We evaluated the overall quality of the evidence through Grading of Recommendations Assessment, Development and Evaluation (GRADE) pro based on the GRADE system. The overall quality of the evidence was classified as high, medium, low, and very low based on the characteristics of the GRADE evidence rating system, such as risk of bias, inconsistency, imprecision, indirectness, and publication bias. A medium overall quality was considered if one of the five items was downgraded, low if two were downgraded, and very low if greater than or equal to three downgrades have been observed.

Data synthesis

Review Manager 5.4.1 software was used for statistical analysis. The relevant outcome index tables of the included studies were all counted. The units and measurement methods were the same, and they were expressed as the mean ± standard deviation, with the mean difference as the effect size. The effect size was considered the 95% confidence interval (CI). The magnitude of heterogeneity between studies was assessed using I2 and P values, and when no statistically significant heterogeneity was observed between studies (I2 < 50%, P > 0.05), a fixed-effects model was used for meta-analysis. If heterogeneity was detected between studies (I2 > 50%, P < 0.05), meta-analysis was performed using a random-effects model, and the total effect value was tested using the Z test, whose results were considered statistically significant if P < 0.05. For the source of heterogeneity in further analysis, if heterogeneity still existed after the analysis, the effect sizes were combined using a random-effects model, and sensitivity analysis was used to test the robustness of meta-analysis results.

Results

Results of the literature search

The PRISMA diagram in Fig. 1 shows the inclusion and exclusion processes for the articles. A total of 2,138 articles, including 234 articles in Chinese and 1,904 articles in English, were screened in this study, and nine articles were finally included. Figure 1 shows the specific screening process.

Basic characteristics of the included literature (Table 1)

Characteristics of participants and intervention

The types of exercise assessed in this review were mainly RE; the interventions mainly included warm-up, stretching and flexion exercises, and RE. The forms of RE comprised sandbag weight-bearing, elastic band, and resistance bicycle exercises. For measurement tools, the Borg scale (n = 7) and exercise heart rate less than 60–70% of maximum heart rate (n = 1) were used. One article literature did not mention how exercise intensity was measured. For the intensity of physical exercise, the participants mainly performed exercise 2–3 times a week for approximately 30–45 min each time; participants in the control group mainly received usual care.

Table 1 Basic characteristics of the included literature.

Studies	Country	Years	Sample	Type of
intervention	Intensity, frequency	Intervention duration	Outcomes	Literature quality score	
I	C	I	C	
Chan et al. (2019)	America	2019	13	15	RE	Usual nursing	70–80% hazard ratio, Borg scale: 12–14 and 2–3/week, and 44 min	12 weeks	①②	3	
Zhang et al. (2020)	China	2020	43	44	Progressive resistance training	Usual nursing	Borg scale: 10–13 and 2–3/week, 1–2 h	12 weeks	①②	4	
Cheema et al. (2007)	Australia	2007	24	25	Resistance Training	Usual nursing	Borg scale: 15–17, 3/week, no indication of duration of each exercise	12 weeks	②⑥	5	
Song & Sohng (2012)	Korea	2012	20	20	Progressive Resistance Training	Usual nursing	Borg scale 11–15, 3/week, 30 min	12 weeks	①⑦⑧	2	
Cai et al. (2022)	China	2022	44	44	REs	Usual nursing	Exercise with heart rate no more than 60–70% of the maximum heart rate, three times a week, 3/week, 45 min	24 weeks	①③④⑤⑥⑨	3	
Zhu (2022)	China	2022	36	35	REs	Usual nursing	Borg scale: 11–13; the frequency and duration of weekly exercise were not stated	24 weeks	①④	3	
Dai & Ma (2021)	China	2020	20	30	REs	Usual nursing	Borg scale: 11–13, 3/week, 40 min	24 weeks	①④⑤⑥⑦⑧⑨	4	
Yan, Zha & Peng (2022)	China	2022	47	47	Progressive RE	Usual nursing	Borg scale: 12–14, 3–4/week, 40–50 min	12 weeks	①④⑤	3	
Tayebi, Ramezani & Kashef (2018)	Iran	2018	17	17	Resistance training	Usual nursing	Intensity of the intervention and the duration of each exercise session were not
stated, and frequency of intervention was 3/week	8 weeks	①⑤⑥	2	
Note:

① Grip strength; ② 6 min walk test; ③ muscle tissue mass; ④ Hb; ⑤ ALB; ⑥ BMI; ⑦ cholesterol; ⑧ low density lipoprotein (LDL); ⑨ urea clear index.

Study design and setting

Table 1 shows the characteristics of the eligible studies 541 individuals participated in this analysis (intervention group: n = 264 and control group: n = 277). This review included studies from six countries around the world, including the USA (n = 1), Australia (n = 1), Korea (n = 1), China (n = 5), and Iran (n = 1).

Literature quality evaluation and risk-of-bias assessment

Nine RCTs were included in this study, and six of them reported in detail how the randomized sequences were generated. Only one study achieved blinding of subjects for allocation concealment and outcome assessment, and the rest eight studies were rated as unclear. Of the nine included studies, one had incomplete data results because its control group, one was rated as high risk because of the significant difference in the number of missed visits in its control and trial groups, and eight were rated as low risk. The risk of bias for each trial was assessed using the Cochrane risk-of-bias tool (Figs. 2 and 3). We also used the Jada scale to assess the quality of the included literature. If the score was less than or equal to three, a study was considered of low quality; a score greater than or equal to four indicated a high quality. The scores are shown in Table 1.

Figure 2 Review authors judgments about each risk of bias item presented as percentages across all included studies.

Figure 3 Reviewer’ judgments about each risk of bias item for each included study.

Publication bias assessment

Egger’s test revealed no significant publication bias between studies (Fig. 4).

Figure 4 Egger’s test plot of grip strength.

Results of GRADE evaluation of outcome indicators (Table 2)

Meta-analysis results

Indexes related to sarcopenia

Grip strength

Changes in grip strength were assessed in seven of nine studies (Tayebi, Ramezani & Kashef, 2018; Yan, Zha & Peng, 2022; Cai et al., 2022; Dai & Ma, 2021; Chan et al., 2019; Song & Sohng, 2012; Zhang et al., 2020) that included 421 patients. A total of 204 patients were assigned to the exercise group and 217 patients to the control group. They were analyzed using a fixed-effects model due to significant heterogeneity (I2 = 0%, P = 0.77). Meta-analysis showed that RE increased the grip strength of patients (MD = 4.39, 95% CI [3.14–5.64], P < 0.00001), with a statistically significant difference (Fig. 5A).

Table 2 Results of GRADE evaluation of outcome indicators.

Certainty assessment	№ of patients	Effect	Certainty	Importance	
№ of studies	Study design	Risk of bias	Inconsistency	Indirectness	Imprecision	Other considerations	RE	Usual care	Relative	Absolute	
(95% CI)	(95% CI)	
Grip strength	
7	RCTs	Serious	Not serious	Not serious	Not serious	None	204	217	-	MD 4.39 higher	⨁⨁⨁◯		
(3.14 higher to 5.64 higher)	Moderate	
The 6-min walk test	
3	RCTs	Serious	Not serious	Not serious	Not serious	None	80	84	-	MD 40.71 higher	⨁⨁⨁◯		
(8.92 higher to 72.49 higher)	Moderate	
Muscle mass	
2	RCTs	Serious	Not serious	Not serious	Not serious	None	64	74	-	MD 4.5 higher	⨁⨁⨁◯		
(2.01 higher to 6.99 higher)	Moderate	
Hb	
4	RCTs	Serious	Serious	Not serious	Serious	None	147	156	-	MD 1.69 higher	⨁◯◯◯		
(1.49 lower to 4.87 higher)	Very low	
ALB	
4	RCTs	Serious	Serious	Not serious	Not serious	None	128	138	-	MD 3.16 higher	⨁⨁◯◯		
(1.13 higher to 5.19 higher)	Low	
Urea clearance index	
2	RCTs	Serious	Not serious	Not serious	Serious	None	44	55	-	MD 0.08 lower	⨁⨁◯◯		
(0.23 lower to 0.07 higher)	Low	
LDL	
2	RCTs	Serious	Not serious	Not serious	Serious	None	40	50	-	MD 1.33 higher	⨁⨁◯◯		
(4.12 lower to 6.77 higher)	Low	
BMI	
3	RCTs	Serious	Very serious	Not serious	Not serious	None	88	99	-	MD 1.46 higher	⨁◯◯◯		
(0.24 lower to 2.67 higher)	Very low	
Cholesterol	
2	RCTs	Serious	Not serious	Not serious	Serious	None	40	50	-	MD 2.33 higher	⨁⨁◯◯		
(5 lower to 9.65 higher)	Low	
Note:

CI, confidence interval; MD, mean difference; SMD, standardized mean difference.

Figure 5 Meta-analysis results for indexes related to sarcopenia.

(A) Grip strength. (B) Six-min walk test. (C) Muscle mass.

Six-min walk test

Among the nine studies, three (Chan et al., 2019; Zhang et al., 2020; Cheema et al., 2007) evaluated changes in the 6 min walk trial involving 164 patients. A total of 80 and 84 patients were assigned to the exercise and control groups, respectively. No heterogeneity was observed between studies (I2 = 42%, P = 0.18). Thus, a fixed-effects model was used for analysis. Meta-analysis showed that RE improved the patients’ ability to walk for 6 min (MD = 40.71, 95% CI [8.92–72.49], P = 0.01), with a statistically significant difference (Fig. 5B).

Muscle mass

Two studies (Cai et al., 2022; Dai & Ma, 2021) reported the effect of RE on muscle mass of 138 patients, that is, 64 patients in the exercise group and 74 in the control group. The heterogeneity test yielded I2 = 0% and P = 0.97, and thus, a fixed-effects model was used for the analysis. The meta-analysis showed that RE increased the muscle mass of patients (MD = 4.50, 95% CI [2.01–6.99], P = 0.0004), with a statistically significant difference (Fig. 5C).

Nutritional indicators

Hb

Exactly four of the nine studies (Zhu, 2022; Yan, Zha & Peng, 2022; Cai et al., 2022; Dai & Ma, 2021) reported changes in Hb levels caused by RE in 303 patients, in which 147 patients were assigned to the exercise group and 156 to the control group. Given the significant heterogeneity (I2 = 69%, P = 0.02), a random-effects model was used for the analysis. Meta-analysis showed that RE did not improve Hb levels in patients (MD = 1.69, 95% CI [−1.49 to 4.87], P = 0.30), and the difference was not statistically significant (Fig. 6A). No significant change was observed in the results after sensitivity analysis, with a case-by-case exclusion of the included studies.

Figure 6 Meta-analysis results for nutritional indicators.

(A) Hb. (B) ALB. (C) BMI. (D) LDL. (E) Cholesterol. (F) Urea clearance index.

ALB

A total of four of nine studies (Tayebi, Ramezani & Kashef, 2018; Yan, Zha & Peng, 2022; Cai et al., 2022; Dai & Ma, 2021) evaluated the effect of RE on ALB in 266 patients. A total of 128 patients were assigned to the exercise group, and 138 were assigned to the control group. A random-effects model was used for the analysis because a significant heterogeneity was observed across studies (I2 = 68%, P = 0.02). Meta-analysis showed that RE improved ALB levels in patients (MD = 3.16, 95% CI [1.13–5.19], P = 0.002), and the difference was statistically significant (Fig. 6B). After sensitivity analysis and exclusion of one study (Dai & Ma, 2021), heterogeneity was reduced (I2 = 4%, P = 0.35), and homogeneity among studies was analyzed using a fixed effects model, which showed that (MD = 4.04, 95% CI [2.44–5.65], P < 0.00001). The meta-analysis results did not change significantly, which indicates their robustness.

Body mass index (BMI)

A total of three out of nine studies (Cai et al., 2022; Dai & Ma, 2021; Cheema et al., 2007) evaluated the changes caused by RE on patients’ BMI; the results of the heterogeneity test were I2 = 82% and P = 0.0004, and thus, a random-effects model was used for the analysis. The findings of meta-analysis showed that RE failed to improve patients’ BMI (MD = 0.44, 95% CI [−2.85 to 3.73], P = 0.79), and the difference was not statistically significant (Fig. 6C). According to the sensitivity analysis, the heterogeneity decreased after excluding one study (Cai et al., 2022) (I2 = 0%, P = 0.65). The studies were homogeneous, and the analysis was performed using a fixed-effects model. The meta-analysis showed insignificant changes (MD = −1.27, 95% CI [−3.29 to 0.75), P = 0.22), which indicated the robustness of results.

LDL

Two articles (Dai & Ma, 2021; Song & Sohng, 2012) reported the effect of RE on LDL. No heterogeneity was observed across studies (I2 = 0%, P = 0.75). Thus, a fixed-effects model was used for analysis. Meta-analysis showed that RE did not improve LDL in patients (MD = 1.33, 95% CI [−4.12 to 6.77), P = 0.63), and the difference was not statistically significant (Fig. 6D).

Cholesterol

Two articles (Dai & Ma, 2021; Song & Sohng, 2012) reported the effect of RE on cholesterol. No heterogeneity was observed across studies (I2 = 0%, P = 0.67), and thus, a fixed-effects model was used for the analysis. The meta-analysis showed that RE did not improve patients’ cholesterol (MD = 2.33, 95% CI [−5.00 to 9.65], P = 0.53), with a non-statistically significant difference (Fig. 6E).

Urea clearance index

Two articles (Dai & Ma, 2021; Cheema et al., 2007) reported the effect of RE on the urea clearance index, and no heterogeneity was observed between them (I2 = 0%, P = 0.66). Therefore, a fixed-effects model was used for the analysis. Meta-analysis showed that RE did not improve dialysis adequacy in patients (MD = −0.08, 95% CI [−0.23 to 0.07], P = 0.29). In addition, the difference was not statistically significant (Fig. 6F).

Discussion

In this study, nine articles were included to analyze the effects of RE on grip strength, muscle mass, 6 min walk distance, Hb level, ALB level, BMI, LDL, cholesterol level, and urea clearance index in patients with MHD, ALB levels, but not BMI, Hb levels, cholesterol, or dialysis adequacy. Thus, RE effectively improves muscle function and strength, but the effect on nutritional status needs to be further validated.

Effect of RE on the indexes related to sarcopenia in MHD patients

In this study, the grip strength index was used to measure the muscle strength of MHD patients, and the results showed that RE resulted in its improvement. Zhang et al. (2020) observed a significant increase in the grip strength of patients who received resistance training in the group with HD, which is consistent with the findings of Neves et al. (2021) and Olvera-Soto et al. (2016). Vogt et al. (2016) showed that grip strength is an independent predictor of all-cause mortality in patients with MHD and is a simple, rapid, and noninvasive measure of muscle strength; in addition, a strong correlation was observed between decreased muscle strength and mortality in dialysis patients. Grip strength can also be used to assess changes in muscle strength. However, it is also an effective indicator for early and rapid identification of malnutrition due to muscle function, and its threshold value for predicting mortality varies by sex and decreases with age. Zhang et al. (2022) also noted that patients with a low grip strength are likely to be at risk of death, and their survival rate is lower than those with higher grip strength values. Several studies have demonstrated (Zhang et al., 2020) that RE effectively improved the grip strength of patients with MHD or those belonging to healthy populations.

RE can increase patients’ muscle strength and mass, promote muscle growth, increase their exercise endurance, and improve their physical function. In this study, RE increased patients’ 6 min walking distance and improved their muscle function. Zhang et al. (2020) and Rosa et al. (2018) also showed that the 6 min walk test is an index commonly used to test patients’ muscle function. In addition, the 6 min walking distance is associated with the quality of survival of HD patients, with a 5.3% increase in life expectancy for every 100 m covered (Zhang et al., 2020; Rosa et al., 2018; Noguchi et al., 2022). However, whether RE increases the exercise capacity of HD patients remains controversial. A meta-analysis of the effect of exercise on HD patients showed that RE did not increase the 6 min walking distance (Huang et al., 2019). By contrast, a meta-analysis by Pu (2018) indicated that RE can increase patients’ 6 min walking distance, which was also shown in the RCT conducted by Zhang et al. (2020) and Rosa et al. (2018).

Therefore, more robust evidence is needed in the future to demonstrate the effect of RE on patients’ 6 min walking distance. A more detailed RE protocol can be specified, and the data can be stratified based on gender, age, dialysis history, and duration of intervention. The content, intensity, and feasibility of RE also need to be considered to reduce patient dropout rates and ensure the integrity and reliability of results.

Effect of RE on relevant nutritional indexes in MHD patients

Patients with CKD receiving HD suffer from malnutrition, which is one of the most common complications in MHD patients, due to the loss of nutrients caused by dialysis and inadequate dietary intake and increased protein energy consumption caused by their disease. Malnutrition adversely affects numerous systems, such as the digestive, cardiovascular, and endocrine systems, in HD patients; its effects include reduced immunity, loss of appetite, renal anemia, infection, atherosclerosis, functional lesions of the heart, brain, lungs, and other organs, and mental fatigue, which result in reduced self-care and quality of life and increased morbidity and mortality in HD patients (Kalantar-Zadeh et al., 2003). The decrease in Hb and serum protein levels increases the risk of these complications in patients (Chen et al., 2021). However, a limited number of studies have been conducted on the effects of RE on the nutritional status of patients with MHD. Zhu (2022) and Yan, Zha & Peng (2022) showed that RE can improve Hb levels in patients, and Gamboa et al. (2020) reported that RE caused improvement on Hb levels. However, Pu (2018) and Dai & Ma (2021) stated that exercise did not improve Hb levels. Therefore, further studies are needed to determine whether RE can improve Hb levels in patients with MHD.

In this study, RE improved serum protein levels in patients with MHD, which reduced the risk of pulmonary infections and improved patients’ quality of life. ALB is a protein required for the body’s nutrition and capillary pressure. It is one of the most commonly used laboratory indicators in evaluating nutritional status (Chen, 2020), and a decrease in its level indicates liver and kidney dysfunction and impaired nutrient absorption in patients (Yan, Zha & Peng, 2022). Most studies revealed that RE increases ALB levels. By contrast, aerobic exercise decreases ALB levels (Bakaloudi et al., 2020); Cheng et al. (2020) observed that after a 2-year RE intervention in dialysis patients, ALB levels were slightly higher. Tayebi, Ramezani & Kashef (2018) and Yan, Zha & Peng (2022) demonstrated that progressive RE can increase ALB levels in dialysis patients, but whether ALB levels can be used as a nutritional indicator to determine the nutritional status of patients remains controversial (Wu, Wang & Wang, 2022).

RE caused no improvement on the BMI. No study has analyzed the effect of RE on BMI. BMI is commonly used to assess whether patients are within the normal weight range, and it is an important influencing factor in the development of pulmonary infections and cardiovascular disease in patients. The level of BMI affects the readmission rate of MHD patients (Tang, 2019), and several studies have shown (Dong, 2017) that MHD patients with obesity and high BMI have a high risk of death; in addition, their survival rates are higher than those of patients with normal or low BMI, which in contrast to the general healthy population. In the present study, RE did not affect BMI values probably because of the short duration of the intervention, and changes in patients’ BMI were unclear.

Abnormal lipid metabolism is relatively common in patients with CKD. This condition is the leading cause of cardiovascular and cerebrovascular development in CKD patients, with elevated LDL and high cholesterol levels being causative factors for cardiovascular disease. However, an observational study of 132 patients undergoing MHD by Zha, Wu & Wang (2017) revealed that high cholesterol and LDL levels in patients were associated with a low mortality. In addition, Liu et al. (2004) showed that cholesterol and LDL levels were negatively associated with mortality. Gordon et al. (2012) conducted a 4-month exercise training program on patients with end-stage renal disease undergoing dialysis after finding that the patients in the exercise group had decreased cholesterol levels. However, in this meta-analysis, RE did not improve the cholesterol levels of dialysis patients. This result may be related to the intensity and frequency of training and the lack of subgroup analysis of lipids.

The urea clearance index is mainly used to determine the changes in the renal function of patients and can be used to assess their nutritional status. This index is commonly used in clinical practice to reflect the dialysis adequacy of patients. An interventional study by Wei (2018) on 100 HD patients revealed that RE improved dialysis adequacy. A reticulated meta-analysis by Luo et al. (2022) showed that aerobic and aerobic exercises combined with RE were the most effective in improving dialysis adequacy relative to RE. The present meta-analysis indicated that RE did not have an effect on dialysis adequacy, which may be related to the small number and sample size of the literature included in this study, low quality of literature, and short duration of intervention prescription and intervention, which did not reflect the effect of RE on the urea clearance index.

Strengths and limitations

Strengths of this study

In this study, we selected RCTs based on RE alone for meta-analysis to validate its effect, verify the effect of RE on muscle strength, muscle function, and walking ability of HD patients, and provide a basis for clinical formulation of the optimal timing of intervention and the optimal frequency, intensity, modality, and content of intervention. In addition, this study provides new evidence regarding the effect of RE on the nutritional status of HD patients. However, the quality of evidence in this study can be improved, and further confirmation of the effect of RE on nutritional status is needed in the future.

Limitations of this study

(1) The literature included in this study showed a high level of heterogeneity, with two articles conducting allocation concealment and subject blinding. (2) The outcome indicators in the literature included in this review were relatively scattered and unfocused, the sample size of the included studies was small, and the trial design of each study was not rigorous. (3) Given the intervention modality of exercise among dialysis patients considered in this study, most of the trials could not be blinded, which resulted in the low quality of the included literature. (4) This review only included intervention studies based on the role of RE alone in improving patients’ physical function and nutritional status, which cannot represent the intervention effects of all exercise modalities. (5) This work did not stratify the intensity and time of exercise intervention implementation, which resulted in a potential bias in the results. (6) In the meta-analysis evaluation of the overall quality of the evidence, most studies had low or deficient levels of evidence; therefore, the results obtained from our meta-analysis cannot be considered conclusive and need to be validated by a large number of studies and a more rigorous trial design.

Implications for future research and practice

This study discusses the effects of resistance exercise on sarcopenia and nutritional status in patients with MHD. First, we integrated the relevant clinical outcomes and analyzed the level of evidence and the shortcomings of resistance exercise applied in clinical trials, which provides a reference for clinical staff to develop more detailed and comprehensive interventions. In future studies of exercise interventions for MHD patients, we may consider stratifying the intensity and frequency of interventions, gender, age, and dialysis history of patients, and extending the duration of interventions to more than 48 weeks to observe changes in clinical outcome indicators, in addition, we need to improve the quality of clinical trials by using proper methods of randomized controlled trials and measures such as allocation concealment and blinding to provide high-quality evidence for the development of future guidelines for prescribing exercise interventions (Liu, Wang & Cao, 2022).

Conclusion

In this meta-analysis, RE increased the grip strength and 6 min walking distance, improved ALB levels, increased muscle strength and function, and enhanced the muscle mass and nutritional status of patients with MHD. However, the effects of RE on Hb levels, cholesterol levels, and urea clearance index of patients with MHD were inconsistent. Several studies have shown that RE had a positive effect on these indexes. However, several research have revealed that RE did not affect these parameters. Therefore, a large number of high-methodology, high-quality studies and more rigorous trial designs are needed in the future to verify the effect of RE on nutritional indexes in patients with MHD.

Supplemental Information

Supplemental Information 1 Raw data.

Click here for additional data file.

Supplemental Information 2 PRISMA Checklist.

Click here for additional data file.

Supplemental Information 3 Details of the search strategies.

Click here for additional data file.

Abbreviations

CKD Chronic kidney disease

MHD Maintenance hemodialysis

HD Hemodialysis

CNKI China National Knowledge Infrastructure

RE Resistance exercise

INPLASY International Platform of Registered Systematic Review and Meta-analysis Protocols

LDL Low-density lipoprotein

MD Mean difference

CI Confidence interval

BMI Body mass index

CBM SinoMed

GRADE Grading of Recommendations Assessment, Development and Evaluation

RE Resistance exercise

CHOLG Cholesterol

ALB Serum albumin

Hb Hemoglobin

Kt/V Dialysis effectiveness index

Additional Information and Declarations

Competing Interests

Author Contributions

Data Availability

The authors declare that they have no competing interests.

Li Li conceived and designed the experiments, performed the experiments, analyzed the data, prepared figures and/or tables, and approved the final draft.

Xiaolan Ma conceived and designed the experiments, performed the experiments, analyzed the data, prepared figures and/or tables, and approved the final draft.

Chunyan Xie performed the experiments, analyzed the data, authored or reviewed drafts of the article, and approved the final draft.

Yamin Li conceived and designed the experiments, authored or reviewed drafts of the article, and approved the final draft.

The following information was supplied regarding data availability:

The raw data is available in the Supplemental File.

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
