# Peer review of "Resistance exercise interventions for sarcopenia and nutritional status of maintenance hemodialysis patients: a meta-analysis"

_PeerJ, doi:10.7717/peerj.16909_

## Round 0.1 · original submission · Minor Revisions

Better explanations about the statistical methods and revisions of the figures presenting the main results are needed.

·

Basic reporting

Sarcopenia is linked to nutritional status and nutritional intervention in HD patients was found to result in increases in muscle mass already. Just alike, nutritional status is considered to have a significant effect on muscle mass, muscle strength, and physical performance.
As per text of this paper I have no significant objections, but_ I think a funnel chart in figure 4 is unnecessary, and as per figs. 6 & 7 – individual effect sizes should be removed, I think that for occasion of publication, forest plots are ok with year & weight only. Additional misgiving is – why are “mean differences” plotted? High P-value and low I2might indicate a problematic sampling (Z values are fine, so the authors are making good point).

Experimental design

As per text of this paper I have no significant objections, but_ I think a funnel chart in figure 4 is unnecessary, and as per figs. 6 & 7 – individual effect sizes should be removed, I think that for occasion of publication, forest plots are ok with year & weight only. Additional misgiving is – why are “mean differences” plotted? High P-value and low I2might indicate a problematic sampling (Z values are fine, so the authors are making good point).

Validity of the findings

n/a

Additional comments

n/a

Reviewer 2 ·

Basic reporting

The manuscript is well-written and clear. I have only one minor suggestion.



Line 37: “However, the improvement caused by RE on hemoglobin and 38 cholesterol levels was not statistically significant.”

--- Recommend reporting actual numbers for these outcomes, as with the rest of the Results section of the abstract. It would be informative for readers as to the numeric improvement caused by RE.

Experimental design

None

Validity of the findings

None

---

## Round 0.2 · accepted · Accept

There are no further comments.

·

Basic reporting

My previous suggestions have beej adressed.

Experimental design

N/A

Validity of the findings

N/A

Reviewer 2 ·

Basic reporting

No additional comment

Experimental design

No additional comment

Validity of the findings

No additional comment